# Comparison of Orthogonal Determination Methods of Acid/Base Constants with Meta-Analysis

**DOI:** 10.3390/ijms252312727

**Published:** 2024-11-27

**Authors:** Tamás Pálla, Károly Mazák, Dania Mohammed Alkhazragee, György Tibor Balogh, Béla Noszál, Arash Mirzahosseini

**Affiliations:** 1Department of Pharmaceutical Chemistry, Semmelweis University, Hőgyes Endre utca 9, 1092 Budapest, Hungary; palla.tamas@semmelweis.hu (T.P.); mazak.karoly@semmelweis.hu (K.M.); dania.khazragee@stud.semmelweis.hu (D.M.A.); balogh.gyorgy.tibor@semmelweis.hu (G.T.B.); noszal.bela@semmelweis.hu (B.N.); 2Center for Pharmacology and Drug Research & Development, Semmelweis University, 1085 Budapest, Hungary

**Keywords:** potentiometry, NMR titration, p*K*_a_ determination, method agreement

## Abstract

The accurate determination of acid/base constants (proton dissociation constants—p*K*_a_, or equivalently protonation constants—log*K*) is essential for the physicochemical characterization of new molecules, especially in drug design and development, as these parameters thoroughly influence the pharmacokinetics and pharmacodynamics of drug action. While pH/potentiometric titration remains the gold standard method for determining acid/base constants, spectroscopic techniques—particularly nuclear magnetic resonance spectroscopy (as NMR/pH titrations)—have emerged as powerful alternatives for specific challenges in analytical chemistry, providing also information on the structure and site of protonation. In this study, we performed a comprehensive meta-analysis of protonation constants reported in the literature, measured using both potentiometry and NMR titrations. Our analysis compiled the available literature data and assessed the agreement between the two methods, taking into consideration various experimental conditions, such as temperature and ionic strength. The results provide insights into the reliability and applicability of NMR titrations compared with potentiometry, offering guidance for selecting appropriate methodologies in drug design.

## 1. Introduction

The acid dissociation constant, *K*_a_, and its negative logarithm, p*K*_a_, are among the most frequently determined physicochemical parameters. Concerning its several applications, it is crucial in the pharmacokinetic properties of a drug, affecting the absorption and thus bioavailability. The definitions related to acid/base processes and pH [1] can be derived from the equilibrium dissociation of a proton (H^+^) from any species, characterized by the acid dissociation constant (*K*_a_):(1)$$K_{a}=\frac{\left[A^{q}\right]\cdot \left[H^{+}\right]}{\left[HA^{q+1}\right]}$$
where the square brackets denote molar concentration, ‘A’ is the species undergoing proton dissociation with charge *q*. Acid/base constants can be determined by several methods, including pH/potentiometry, which is considered to be the standard one. Other pH titrations combined with a spectrometric (e.g., UV, CD, IR, or NMR) or a separation technique (e.g., HPLC, electrophoresis, or CE), or electrochemical methods (e.g., conductometry or voltammetry), can also be used [2]. Potentiometry is based on the measurement of the activity of the hydrogen ions (protons) in a solution of known acid/base ratio of the compounds, and the pH is usually monitored by a combined glass electrode [3]. Potentiometry is useful for p*K*_a_ determination (usually in the 2–12 pH range) in aqueous media if the compound is soluble and stable; note that in this work we focus only on aqueous media; methods using cosolvents are not discussed. If solubility/sample purity does not allow for potentiometry, pH titration with spectrometric methods that are more sensitive/afford selective signals on the ionizable group may be preferential [4].

NMR/pH titrations for the determination of acid/base constants have the benefit (compared with most spectroscopy titrations) of high selectivity in the spectrum; i.e., with sufficient measurement time and resolution, the signals of the compound of interest can be observed even if impurities are present in the sample, as long as at least one reporter signal of the analyte can be observed throughout the titration. The high resolution in NMR potentially also affords atom-specific information on the degree of perturbation caused by ionization, or even protonation-induced conformational changes can also be inferred [5]. The obvious drawback of NMR-based titrations is the expensive instrumentation that is required, compared with the fast, inexpensive, and automated methods of potentiometry and even UV spectroscopy-based titrations. A comparison of practical aspects of pH/potentiometry and NMR/pH titrations is presented in Table 1 [6]. For many cases, it has been reported that there is excellent agreement between NMR- and potentiometry-based values [7]; however, in this work we aim to extend the systematic literature review of available acid dissociation constants and perform a statistically sound analysis of agreement between the two analytical methods.

## 2. Materials and Methods

### 2.1. Data Collection

Data for this study were systematically collected from existing literature using a comprehensive search strategy. We utilized multiple academic databases, e.g., Reaxys and PubChem (including the database of Critical Stability Constants [9,10]), to identify relevant peer-reviewed articles, conference papers, and reviews. Inclusion criteria were established to select studies that provided quantitative data with acceptable quality pertinent to our research questions. Inclusion criteria were the appropriate description of experimental design and method from which the exact method of determination could be inferred. Unfortunately, many instances of literature papers do not report the uncertainty of determination (standard error or standard deviation); however, in the mixed-effects model, the standard error of the values was not incorporated directly in the model, since it is apparent from the distribution standard errors that an appropriate variance weight function can be used, i.e., a lack of standard error was not considered as an exclusion criterion. The collected data were screened independently by two researchers to confirm eligibility. The variables recorded from the literature were the name of the compound, the value of the acid/base constant, the standard error of measurement (where available), and two experimental conditions: temperature and ionic strength. All values used pertain to aqueous (or, in the case of NMR measurements, 5–10% D_2_O) solutions.

### 2.2. Statistical Analysis and Mathematical Calculations

The data analysis was conducted using the R programming language (R version 4.0.5 (R Foundation for Statistical Computing, Vienna, Austria) [11] and the R Studio integrated development environment (Posit, Boston, MA, USA). Visualization of the results was performed using R Studio software (version 30.4.0). Bland–Altman analysis was performed based on the original publication [12]. The mixed-effects model was built using the ‘nlme’ package [13], while cheminformatics data were manipulated using the ‘ChemmineR’ package [14]. See the R script in the Supporting Information for all packages and functions used.

## 3. Results

The observed variables and the full dataset can be found in the Supporting Information section. The compiled data table contains 417 records of 112 compounds coming from 43 studies [7,15,16,17,18,19,20,21,22,23,24,25,26,27,28,29,30,31,32,33,34,35,36,37,38,39,40,41,42,43,44,45,46,47,48,49,50,51,52,53,54,55,56]; 125 records had standard error values missing, while 74% of measurements were conducted at 25 °C and 0.15 mol/L ionic strength (which is considered the standard condition for potentiometric titrations). Potentiometric titrations were difference titrations (carried out in the absence and presence of the analyte), while NMR titrations utilized ^1^H reporter signals. A sample section of the data table can be found in Table 2 with the example of oxidized glutathione (glutathione disulfide, GSSG).

The pairs plot of selected variables is shown in Figure 1, revealing relationships or lack thereof between some variables. Notably, there seems to be considerable difference neither in location nor in scale between p*K*_a_ values and their standard errors among the two methods (see plots in column 4, rows 2 and 3).

The observed values of p*K*_a_ are depicted in a histogram in Figure 2 to reveal the distribution of acid dissociation constants that are present in the dataset. The standard errors of the acid dissociation constants reported in the literature are also shown on a histogram to reveal an exponential distribution. Note that the practical lower limit of p*K*_a_ determination is on the order of 0.01 (the most common value in the dataset) due to the capability of the combined glass electrode [55].

The Bland–Altman plot of p*K*_a_ values is shown in Figure 3 to reveal the difference between the two methods of determination. The differences between NMR-determined values and potentiometry-determined values are shown for each p*K*_a_ on the y-axis, and the mean of the determined values by the two methods is shown on the x-axis. Note that the difference (NMR−Pot) is merely performed in alphabetical order and is not meant to represent NMR being the reference method from which the deviance of potentiometry is observed, i.e., in this model both methods are treated as being estimates of the same true value. Were the opposite differences (i.e., Pot−NMR) calculated, the Bland–Altman plot would be the same only inverted on the x-axis. The solid red line is the mean of all differences, which should lie on the 0 horizontal line (dashed grey line) if the methods are in complete agreement. However, a deviation of the mean difference (towards the positive in this case) shows signs of bias, which in this case translated to NMR-determined p*K*_a_ values tending to be on average somewhat greater (by 0.056 log units) than the potentiometry-determined values of the same proton association step. Dashed red lines show the 95% limits of agreement of the difference values, calculated from the variance of the data points. Two methods are considered to be in agreement (i.e., can be used interchangeably) if the limits of agreement fall within scientifically relevant/important bounds of equivalence. This bound is determined a priori; for acid/base titrations with conventional methods, a scientifically accepted value is 0.2 log units [57] and is shown with solid blue lines for reference on the Bland–Altman plot (Figure 3). Bias is indicative of systematic error, while variance is characteristic of random error. The Bland–Altman plot also reveals some serious outliers, typically when extreme p*K*_a_ values were measured; therefore, this analysis was repeated after these outliers were trimmed from the data. It is also noteworthy that the most commonly used method for measuring method agreement, correlation of the measured value with the two methods, is not recommended by statisticians, since it can be misleading and measures the degree of association, not agreement.

After removing the outlier data points as shown in Figure 3, an updated Bland–Altman plot is achieved in Figure 4 with somewhat lower bias (0.042 log units) and standard deviation of difference values. In the updated Bland–Altman plot, three variants are shown. In the first case we also sought to depict p*K*_a_ values that stem from overlapping dissociation steps with red circles, while non-overlapping p*K*_a_ values (or monoprotic values) are with black circles, in order to assess whether overlapping proton dissociation causes any bias between the two methods. On visual inspection, there seems to be no greater bias on the p*K*_a_ values of overlapping proton dissociation. However, a slight trend is observed with greater positive bias associated with higher p*K*_a_ values, and vice versa, more negative bias is associated with lower p*K*_a_ values. A simple linear regression performed on the trimmed Bland–Altman data reveals a slope significantly different from 0 (*p*-value: < 0.0001), albeit with a rather weak degree of association (slope: 0.019, std. error: 0.004, adj. R^2^: 0.101). In the second variant, the data points are colored depending on whether the moiety in question has acidic or basic character (i.e., upon dissociation, the charge of the moiety changes from neutral to negative—acidic moiety—or from positive to neutral—basic moiety). Amphoteric character is assigned to p*K*_a_ values where overlapping dissociation of two moieties with opposite acid/base character occurs (e.g., in morphine); note that these values could only be unequivocally assigned to a single moiety if the values were microscopic constants. It can be seen that mostly the different acid/base characters cluster on the two ends of the p*K*_a_ scale, and basic moieties have a much wider scattering. It is also noteworthy that the biases of acidic and basic moieties are close to the 0 horizontal line (positioned on the opposite sides), and thus it is the bias attributed to mixed moieties (and the fact that there are more acidic moieties in the dataset) that causes the overall bias towards NMR. On the other hand, the scattering of mixed moieties is the smallest. In the third variant of the Bland–Altman plot, data points are colored according to the method of pH determination during NMR/pH titrations (as this was different among the literature). The highly accurate in situ pH indicators are considered to be reliable even at extreme pH values, while sometimes the conventional combined glass electrode is used for pH measurement before the NMR spectrum recording. It is apparent from this figure that most of the variance with indicator pH measurement occurs at the two ends of the p*K*_a_ scale (below 2.5 and above 8.5). On both sides of the p*K*_a_ scale, this variance is directed upwards from the 0 horizontal line, resulting in a larger bias towards NMR compared with the overall dataset. The bias of electrode pH determination is very close to zero (as would be expected, since both methods now use the same principle for pH determination); however, the linear trend of differences vs. the expected value of p*K*_a_ is now more pronounced (slope: 0.027, std. error: 0.008, adj. R^2^: 0.13).

In order to examine whether the difference between the two methods follows a normal distribution, the histogram and Q-Q plot of difference values are depicted in Figure 5. Both plots show a somewhat left-skewed distribution; however, the histogram appears symmetrically positioned at zero value, with lower extreme values being somewhat more frequent than extreme larger values.

In order to quantify the effect of the determination method, taking into account the correlating nature of p*K*_a_ values within the same compound, a mixed-effects model was fitted to the entire dataset (without trimming) with the following formula:(2)$$pK_{a,ij}=\left(\beta_{0}+u_{0}\right)+(\beta_{1}+u_{i}+u_{ij})\cdot {method}_{ij}+\epsilon_{ij}$$

The above (original) mixed-effects model assumes a correlation structure between the acid/base constants within the same compound in a random intercept setting: p*K*_a,*ij*_ is the determined value of compound *i* at step *j*; *β*_0_ is the mean value of NMR derived values; *u*_0_ is the mean deviation of the NMR derived values of compound *i*; *β*_1_ is the mean effect of potentiometry; *u_i_* is the mean deviation of the effect of potentiometry on compound *i*; *u_ij_* is the mean deviation of the effect of potentiometry on protonation step *j* of compound *i*; *method_ij_* is the indicator variable (0 for potentiometry and 1 for NMR, i.e., potentiometry is considered as the reference level to be congruent with the director of differences used in the Bland–Altman analysis); *ϵ_ij_* is the random error term. The matrix *u* = (*u*_0_, *u_i_*, *u_ij_*)^T^ is assumed to have a multivariate normal distribution with a mean vector of (0, 0, 0)^T^ and covariance matrix *G* and is independent from *ϵ_ij_,* which is assumed to have a normal distribution with a mean value of 0 and variance *σ*^2^. This original model affords a 0.062 coefficient for the difference between potentiometry and NMR methods (close to the 0.056 difference in the Bland–Altman plot for the NMR/potentiometry method difference), and this difference is significant at the 5% level (*p*-value: 0.0044). The above mixed-effects model was then extended to model the variance structure, i.e., taking into account that the standard error of p*K*_a_ values depends on the p*K*_a_ value itself (hence the outliers observed in Figure 3), an observation that can also be made in the pairs plot of Figure 1: column 2, row 3, the plot shows an increasing standard error for extremely low and extremely high p*K*_a_ values, which is characteristic of combined glass electrode measurement error. In the extended model, therefore, an exponential variance weight function was applied. The output of the above two models is presented in Table 3. The extended model affords a 0.041 coefficient for the difference between potentiometry and NMR methods (in agreement with the 0.042 difference in the Bland–Altman plot, Figure 4), and this difference is now barely significant at the 5% level (*p*-value: 0.0512). The diagnostic figures of the two models are shown in Figure 6, revealing a slight improvement in variance homogeneity in the extended model. The likelihood-ratio ANOVA test between the two models shows a significant difference (likelihood-ratio test: 7.1636, *p*-value: 0.0074).

In order to gauge the effect of measurement parameters (ionic strength, temperature), the above mixed-effects models were extended in their fixed effects by these variables in interaction with the titration method. No significant effects were observed for any interaction terms, and then, based on standard model selection procedures, the non-significant interaction terms followed by the main effect terms were removed until no significant effects remained from ionic strength and temperature. Furthermore, in the Bland–Altman plot of the trimmed data, it was examined whether data points were affected by a differing temperature or ionic strength condition when determined with the two different titration methods; however, the number of such data was small, and there was no observable trend with respect to this comparison. These results can be reproduced using the R code in the Supporting Information.

In order to assess the heterogeneity of the molecular structures in the dataset, the occurrence of functional groups (listed in Table 4) and atom frequency (Figure 7) were quantified. The structural compound descriptors were then clustered based on their pairwise distances. This results in a multi-dimensional scaling plot similar to hierarchical clustering visualized in Figure 7. The distribution of acidic/basic/mixed moieties was 76, 42, and 46 (46%, 26%, and 28%). The Tanimoto coefficient of the dataset was measured against the FDA-approved drugs, which afforded 18%.

## 4. Discussion

The comparison of protonation, or equivalently, acid dissociation constants (p*K*_a_ values) obtained by pH/potentiometry and NMR/pH has revealed important insights into the performance and bias of each method. The analysis of 425 records from 114 compounds demonstrated that these methods provide comparable results; some systematic and random errors, however, persist, particularly for compounds of extreme p*K*_a_ values.

### 4.1. Agreement Between Methods

The Bland–Altman analysis (Figure 3) revealed a small but systematic bias, with NMR-determined p*K*_a_ values tending to be higher than those obtained via potentiometry, with an average difference of 0.056 log units. This bias implies that NMR may slightly overestimate or potentiometry may underestimate acid/base constants. Importantly, the standard deviation of differences also indicated a degree of random error between the methods, which was more pronounced in certain cases, such as the catecholamines, where outlier behavior was observed.

Upon removal of outliers (Figure 4), the bias was reduced to 0.042 log units, and the variance of differences also decreased, confirming that extreme p*K*_a_ values—particularly those associated with certain chemical classes—had a significant impact on the observed differences between the two methods. One class of chemical moieties that suffers from measurement error during potentiometry is the phenolic hydroxyl group in catecholamines, as they are very sensitive to both oxidation and alkaline pH. While the decomposition or chemical transformation of sensitive compounds hinders potentiometric titrations, NMR measurements remain unaffected as long as the reported signals of the analyte can be monitored. A weak but statistically significant linear trend found after outlier removal suggests that larger p*K*_a_ values tend to show more positive bias, while smaller p*K*_a_ values exhibit a negative bias; this trend is more pronounced if the pH was measured during NMR titrations with a combined glass electrode (the same as in pH/potentiometric titrations). This indicates that the magnitude of p*K*_a_ might influence the direction of bias between NMR and potentiometry if the method of pH determination is identical. When in situ pH indicators are used, this trend becomes less apparent; however, the large scattering of method differences at the p*K*_a_ extremes still remains. One source of the apparent trend in the differences could be the clustering of acidic moieties on the lower p*K*_a_ range, while basic moieties lie on the higher end of the p*K*_a_ scale. Another obvious source of method disagreement is the acidic/alkaline error of glass electrodes. Due to the many factors affecting the accuracy of pH readings with glass electrodes, it is apparent that a controlled environment (e.g., temperature control and inert atmosphere) is crucial for pH/potentiometric titrations in order to show good agreement with other methods, such as NMR/pH titrations with in situ indicators, where the temperature-controlled environment in the NMR probe is a given. One other source of the trend in method differences vs. p*K*_a_ values could be a statistical artifact. When a simulation is performed to model the measurement uncertainties inherent to the two methods in question, it is clear that when fitting a non-linear curve to a titration dataset in which the ends of the pH scale are burdened with greater uncertainty compared with the intermediate pH range, the estimates of the regression will overestimate low p*K*_a_ values and underestimate high p*K*_a_ values. In the simulation, the uncertainty pertaining to chemical shift readings and volume differences in NMR and potentiometry titrations, respectively, was modeled by adding noise sampled from normal distributions of different variances. The results of the simulation are presented in Figure 8; the R code of the simulation can be found in the Supporting Information.

### 4.2. Error Distribution and Methodological Differences

Histograms and Q-Q plots of the difference values (Figure 5) showed a left-skewed distribution, although the differences appeared symmetrically centered around zero. This observation, combined with the Bland–Altman plots, indicates that while the methods are generally in agreement, NMR may introduce slightly larger errors, especially at the extremes of the p*K*_a_ range. The skewed nature of the distribution could be attributed to measurement noise or instrument limitations, particularly in potentiometry, where combined glass electrode accuracy may degrade at very low or high p*K*_a_ values.

Moreover, the exponential distribution of the standard errors (Figure 2) aligns with the expectation that measurement errors increase at extreme p*K*_a_ values. This was supported by the pairs plot (Figure 1), where the increase in standard error at both ends of the p*K*_a_ scale was evident. Such behavior is characteristic of potentiometric measurements due to electrode limitations, suggesting that at very low or high p*K*_a_ values, NMR may outperform potentiometry in precision, despite the observed bias. Upon inspection of the p*K*_a_ errors reported, grouped by acid/base moieties, no considerable differences were observed (see Supporting Information).

### 4.3. Mixed-Effects Models

The mixed-effects models provided a more robust framework for assessing the impact of method choice on p*K*_a_ determination while accounting for the inherent correlation between multiple proton dissociation steps within the same compound. The original model estimated a significant (albeit scientifically unimportant) difference between the two methods (0.062 log units, *p*-value: 0.0044), consistent with the findings of the Bland–Altman plot. However, after adjusting for variance heterogeneity (which was particularly evident in compounds with extreme p*K*_a_ values), the extended mixed-effects model yielded a smaller and non-significant difference (0.041 log units, *p*-value: 0.0512). This suggests that much of the bias can be attributed to variance-related issues in the data, particularly those arising from potentiometric measurement errors at extreme p*K*_a_ values.

### 4.4. Implications and Limitations

While both NMR and potentiometry are widely accepted methods to determine acid/base constants, this study highlights subtle differences in their performance, particularly when analyzing compounds with very high or low p*K*_a_ values. This comparison extends the work conducted by Bezençon et al. [7], where these two methods were compared by regression and correlation analysis only; the conclusions remain the same; however, as good agreement is found between the two methods. The overestimation of p*K*_a_ by NMR could be due to method-specific factors such as its sensitivity to overlapping proton dissociation steps, as suggested by the visual inspection of the Bland–Altman plots (Figure 4). However, no significant bias was observed for overlapping proton dissociation steps when these were isolated in the trimmed dataset.

Additionally, no significant interaction effects were observed between the method and measurement parameters such as temperature or ionic strength, suggesting that the bias between methods is relatively stable across different experimental conditions. Nonetheless, the weak correlation between bias and p*K*_a_ values suggests that further refinement in measurement techniques could help minimize these discrepancies. Finally, while outliers were identified and removed in the Bland–Altman analysis, the presence of such extreme values—particularly in catecholamines—indicates that careful attention should be paid to certain chemical classes that may introduce greater variability in p*K*_a_ determination. For instance, the outliers observable in Figure 3 reveal that extremely basic moieties like the fully deprotonated catechol ring are prone not only to the alkaline error but are also exposed to oxidation by air; therefore, potentiometric titration is a suboptimal choice. Other groups that are easily oxidized (e.g., selenols and thiols in selenocysteine or cysteine) are also sensitive to air, and therefore NMR is more suitable to track the reporter protons of the analyte, excluding therefore the perturbing effects of the oxidation product. Other characteristic extreme p*K*_a_ values pertaining to strong carboxylic acids or strong basic moieties (e.g., the guanidino group in arginine or amino groups in polyamines) require careful consideration of the method chosen for p*K*_a_ determination. Future studies could benefit from exploring the molecular properties that contribute to such outliers and developing method-specific correction factors or adjustments.

## 5. Conclusions

NMR and potentiometry generally provide consistent acid/base constant measurements; the systematic slight overestimation of NMR-derived p*K*_a_ values compared with pH/potentiometry is not scientifically relevant; however, the large variance between the two methods results in limits of agreement exceeding the bounds of what would be methodologically acceptable (these limits of agreement come close to the acceptable bounds when only NMR titrations with in situ pH indicators are considered). This bias is further diminished when accounting for variance structures, particularly in compounds with extreme p*K*_a_ values. No significant effects of experimental conditions (temperature, ionic strength) were observed, suggesting that these methods are sufficiently robust across a range of experimental setups. Future work should focus on reducing measurement errors, particularly for compounds with extreme p*K*_a_ values, to further harmonize results between these two methods. With these insights in mind, NMR remains a powerful tool for determining acid dissociation/protonation constants, even when high-accuracy data are required. In fact, outside the p*K*_a_ range of 2–10, the use of NMR titration with in situ pH indicators is preferable. In the intermediate p*K*_a_ range, the two methods can be used interchangeably.

## Figures and Tables

**Figure 1 ijms-25-12727-f001:**
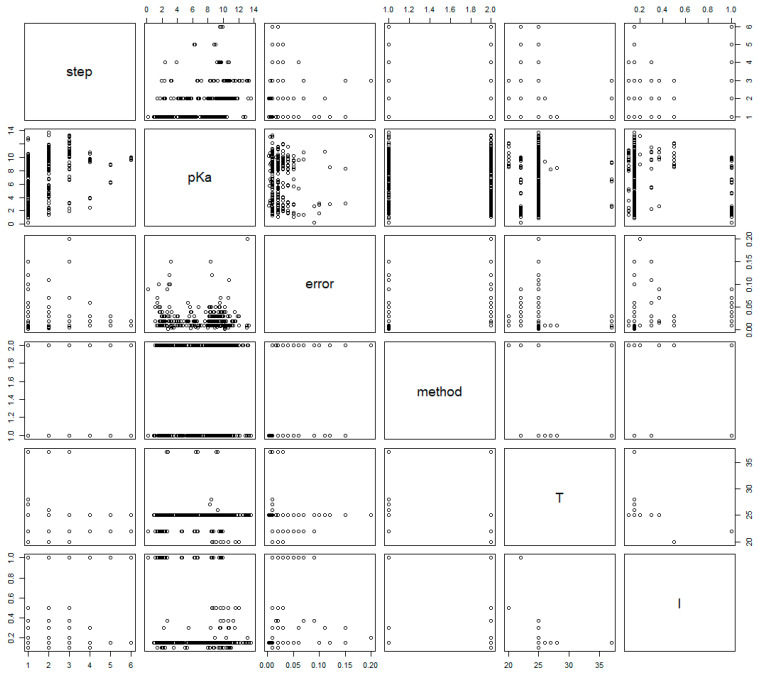
Pairs plot of selected variables. Note that methods ‘1’ and ‘2’ are used to denote ‘NMR’ and ‘Pot,’ respectively.

**Figure 2 ijms-25-12727-f002:**
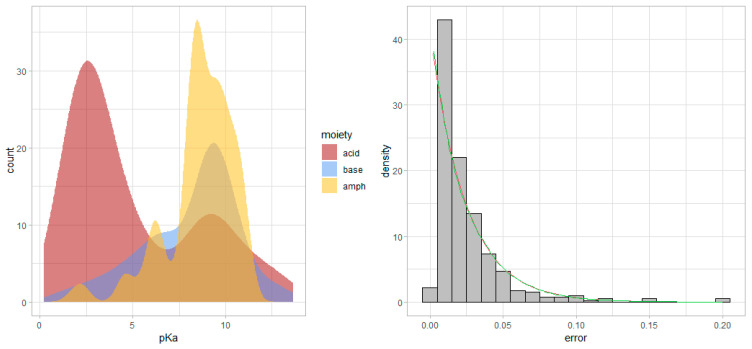
**Left**: the smoothed frequency histogram of p*K*_a_ values depicted for the acid/base/amphoteric moieties, with a multimodal empirical distribution. **Right**: the density histogram of the standard error of p*K*_a_ values is an exponential empirical distribution; the exponential theoretical distribution calculated from the mean (red line) and standard deviation (green line) of the error values, respectively, are also shown.

**Figure 3 ijms-25-12727-f003:**
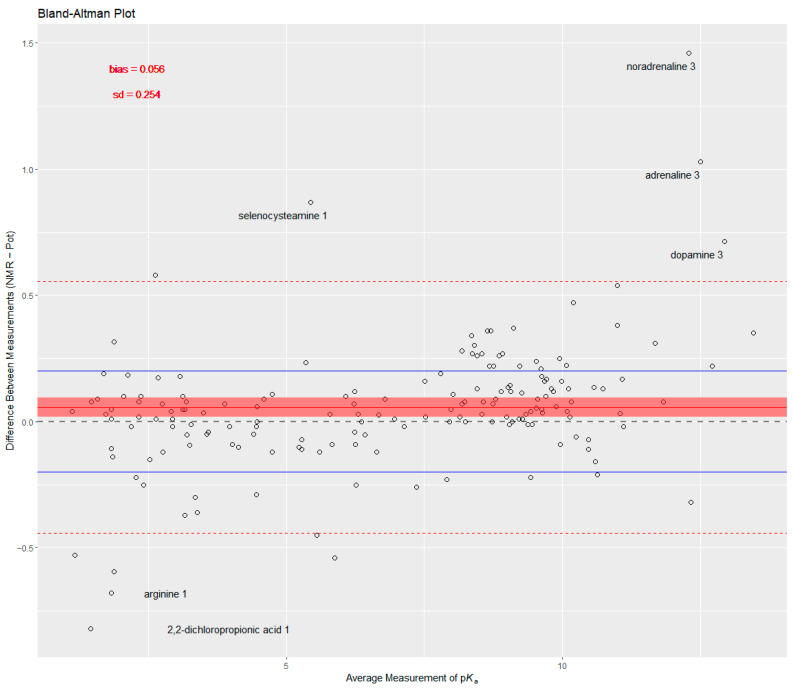
The Bland–Altman plot of p*K*_a_ values determined with the two methods: potentiometry and NMR; on the y-axis, the difference: NMR-determined values minus potentiometry-determined values. The outlier p*K*_a_ values are identified with text labels (compound and protonation step). The 95% limits of agreement are at dashed lines, while the 95% confidence interval of the bias is depicted with shaded areas. Bounds of scientifically important differences are shown with solid blue lines.

**Figure 4 ijms-25-12727-f004:**
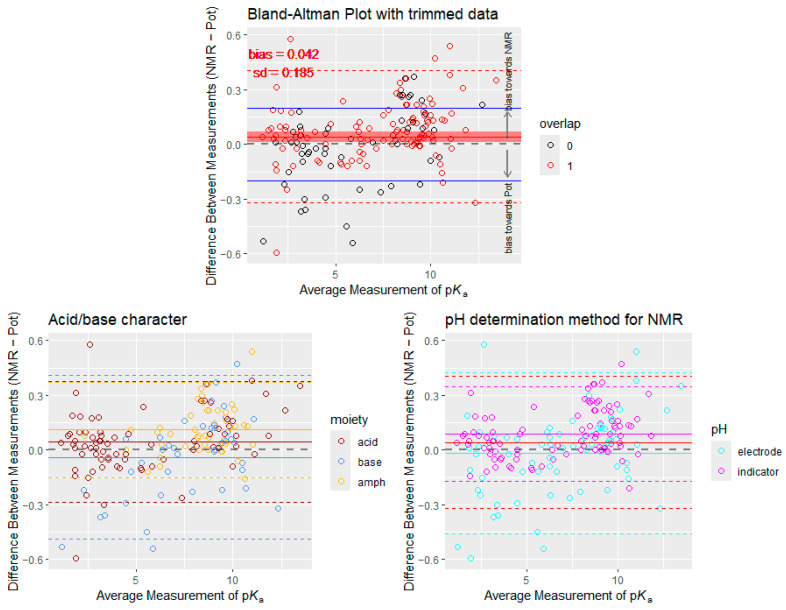
The Bland–Altman plot of p*K*_a_ values with outliers removed. On the top, values from overlapping protonation are shown with red circles; two successive protonation steps were considered overlapping if their difference was below 2 p*K*_a_ units. The bias and 95% limits of agreement of the entire dataset are shown in red for reference, together with the bounds of scientifically important differences with solid blue lines. On the bottom left, values from the acidic/basic/amphoteric moieties are shown in color together with their bias and 95% limits of agreement. On the bottom right, values of the two pH measurement techniques during NMR titrations are shown in color together with their bias and 95% limits of agreement. The bias and 95% limits of agreement of the entire dataset are shown in red for reference.

**Figure 5 ijms-25-12727-f005:**
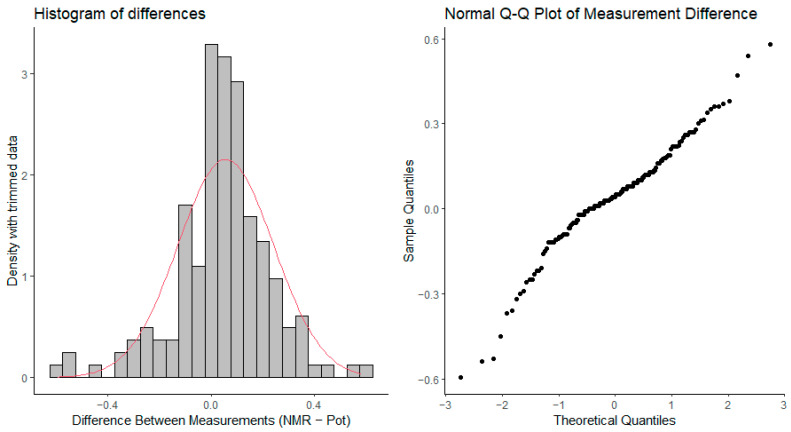
Left: the density histogram of p*K*_a_ difference values from the trimmed data of Figure 4, together with the theoretical normal distribution of the same mean and variance in red. Right: the Q-Q plot of p*K*_a_ difference values from the trimmed data.

**Figure 6 ijms-25-12727-f006:**
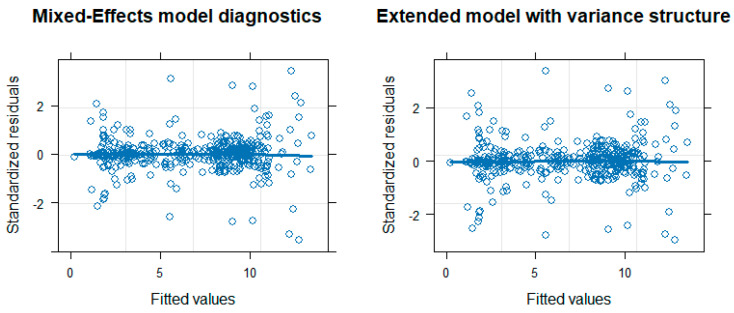
Diagnostic figure of the original mixed-effects model (**left**) and the extended mixed-effects model (**right**).

**Figure 7 ijms-25-12727-f007:**
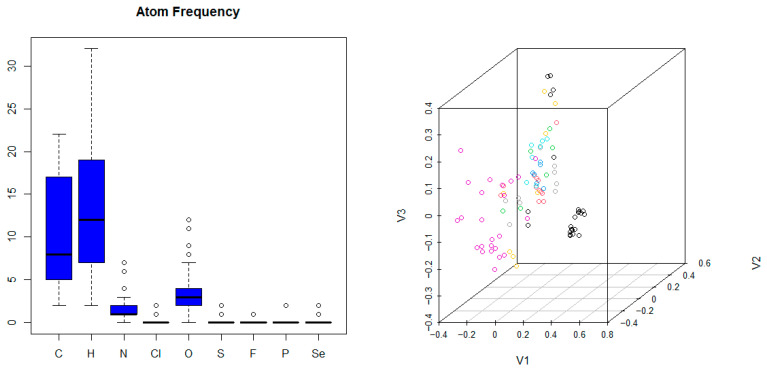
On the left: the atom frequency in the compounds of the dataset. On the right: the multi-dimensional scaling plot of the compounds after clustering using the Tanimoto distances.

**Figure 8 ijms-25-12727-f008:**
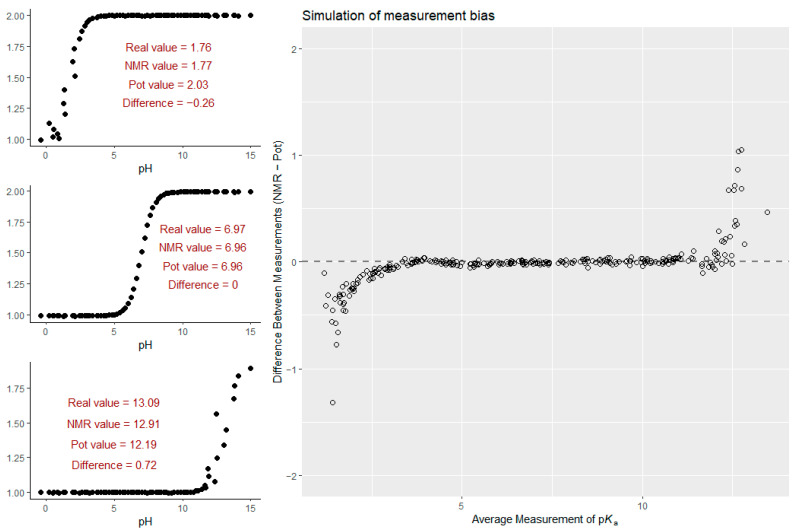
On the **left**, three representative NMR titration curves with simulated low, medium, and high pK_a_ values; the simulated value is depicted as the ’real’ value, and after modeling the measurement uncertainties inherent to both methods, the fitted values and their difference are also depicted. On the **right**, a simulation of 300 pK_a_ values from a uniform distribution afforded a Bland–Altman plot, in which the bias trend vs. pK_a_ can be observed.

**Table 1 ijms-25-12727-t001:** Comparison of potentiometry and NMR spectroscopy-based determination of acid/base constants in aqueous solutions.

	Potentiometry	NMR/pH Titration
Principle	Measurement of electrode potential at each degree of titration	Measurement of (usually) chemical shifts, which change with pH
Accuracy	Very accurate, especially for systems with clear ion selectivity; inert atmosphere required but limited mainly to the 2–12 pH range	Very accurate, especially for complex molecules. Inert atmosphere can be avoided, and detectable pH range can be extended with in situ pH indicators [8]
Complexity	Simple; requires calibration and maintenance of the electrodes	More complex NMR measurements needed. Isotope effect of the solvent may need to be accounted for
Sample requirements	Sample must be pure (>95%) and of sufficient concentration (0.1–10 mmol/L)	Sample must be in sufficient concentration (>1 mmol/L) for NMR detection (some impurities may be present)
Time, efficiency, throughput	Fast, especially for automated instruments, medium throughput	Slow, due to multiple scans and data interpretations. Low throughput; can be made faster and medium throughput using autosampler
Cost	Low cost, simple instruments	Expensive, due to high equipment cost and maintenance
Titration of multiprotic molecules	Characterizes the molecule as a whole, in the case of overlapping protonation steps as well	Provides information on specific protonation sites

**Table 2 ijms-25-12727-t002:** Example from the compiled data table (restructured to wide arrangement) of acid/base constants determined with both pH/potentiometry and NMR/pH titration.

Compound	Step	NMR p*K*_a_	Pot. p*K*_a_	NMR Std. Err.	Pot. Std. Err.	Study	*t* (°C)	*I* (mol/L)
oxidized glutathione	1	1.79	1.6	±0.02	±0.1	[37]	25	0.15
oxidized glutathione	2	2.42	2.32	±0.02	±0.03	[37]	25	0.15
oxidized glutathione	3	3.23	3.15	±0.02	±0.02	[37]	25	0.15
oxidized glutathione	4	3.92	3.85	±0.02	±0.01	[37]	25	0.15
oxidized glutathione	5	8.95	8.83	±0.02	±0.01	[37]	25	0.15
oxidized glutathione	6	9.71	9.53	±0.01	±0.02	[37]	25	0.15

**Table 3 ijms-25-12727-t003:** The output of the original and extended mixed-effects models.

Original Mixed-Effects Model	Extended Mixed-Effects Model with Variance Structure
AIC = 1098.627	AIC = 1093.463
BIC = 1118.659	BIC = 1117.501
LogLik = –544.3134	LogLik = –540.7316
Random effects:	Random effects:
Formula: ~1|compound	Formula: ~1|compound
intercept sd: 1.313	intercept sd: 1.313
Formula: ~1|step in compound	Formula: ~1|step in compound
intercept sd: 3.031	intercept sd: 3.032
residual sd: 0.209	residual sd: 0.1614
Fixed effects (value, std.error, t-value):	Fixed effects (value, std.error, t-value):
intercept:	Intercept:
6.66, 0.26, 25.8669	6.67, 0.26, 25.8959
Effect of method (NMR):	Effect of method (NMR):
0.062, 0.022, 2.8751	0.041, 0.021, 1.9611
Degrees of freedom = 213	Degrees of freedom = 213
Correlation of intercept~NMR: –0.042	Correlation of intercept~NMR: –0.04 Variance parameter estimate: 0.0339
ANOVA (F-value, *p*-value)	ANOVA (F-value, *p*-value)
intercept:	intercept:
676.5005, <0.0001	675.7902, <0.0001
method:	method:
8.2662, 0.0044	3.8461, 0.0512

**Table 4 ijms-25-12727-t004:** The occurrence (in percentage) of functional groups and rings in the compounds of the dataset.

RNH_2_	R_2_NH	R_3_N	ROH	RCOR	RCOOH	RCOOR	ROR	RCCH	Rings	Arom
30	19	26	34	11	49	10	24	1	61	59

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
