# Peer review of "Comparison of Orthogonal Determination Methods of Acid/Base Constants with Meta-Analysis"

_ijms, 2024, doi:10.3390/ijms252312727_

Round 1

Reviewer 1 Report

Comments and Suggestions for Authors

The authors present a comparative meta-analysis comparing the measurement of protonation constants for organic compounds as reported in scientific literature, utilizing potentiometry and NMR titrations. The findings indicate that while NMR tends to slightly overestimate pKa values compared to potentiometry, this overestimation is not of significant scientific concern.

The study's objectives are clearly stated, and the methodology is robust. This analysis effectively addresses a potential issue faced by researchers and elucidates several key points. Indeed, some might consider one methodology more practical than the other, and this paper provides a strong background to assist researchers in justifying their choice.

Minor comments

- My main concern pertains to the journal's scope: IJMS primarily focuses on research in molecular biology and medicinal chemistry. While this research is rooted in analytical chemistry, it remains highly relevant to medicinal chemistry. Therefore, I recommend that the authors offer additional details regarding the types of molecules and their functional groups and comment on how the different structures and their chemical and physical properties might affect the measurement of pKa on the basis of the results of some of the experimental works considered for this meta-analysis.

- Table 2 should be rearranged by shifting the Potentiometry measurement values in a column near the NMR measurement values and reporting the Std.error as ± x

Author Response

Comment 1:

My main concern pertains to the journal's scope: IJMS primarily focuses on research in molecular biology and medicinal chemistry. While this research is rooted in analytical chemistry, it remains highly relevant to medicinal chemistry. Therefore, I recommend that the authors offer additional details regarding the types of molecules and their functional groups and comment on how the different structures and their chemical and physical properties might affect the measurement of pKa on the basis of the results of some of the experimental works considered for this meta-analysis.

Response 1:

The utility of the performed meta-analysis was supported in the manuscript by assessing the heterogeneity of the molecular structures in the dataset e.g. with the ocurrence of functional groups. The Tanimoto coefficient of the dataset was measured against the FDA approved drugs which afforded 18%. Furthermore the identification of outlier pKa moieties in the dataset could help guide decision in chosing a method of determination, thus the following was added to the Discussion:

"For instance, the outliers observable in Figure 3 reveal that extremely basic moie-369 ties like the fully deprotonated catechol ring is prone not only to the alkaline error but is 370 also exposed to oxidation by air, therefore potentiometric titration is a suboptimal choice. 371 Other groups that are easily oxidized (e.g. selenols and thiols in selenocysteine or cyste-372 ine) are also sensitive to air and therefore NMR is more suitable to track the reporter pro-373 tons of the analyte, excluding therefore the perturbing effects of the oxidation product. 374 Other characteristic extreme pKa values pertaining to strong carboxylic acids or strong 375 basic moieties (e.g. guanidino group in arginine or amino groups in polyamines) require 376 careful consideration of the method chosen for pKa determination."

Comment 2:

Table 2 should be rearranged by shifting the Potentiometry measurement values in a column near the NMR measurement values and reporting the Std.error as “± x”

Response 2:

The suggested changes were made to the manuscript.

Reviewer 2 Report

Comments and Suggestions for Authors

The paper deals with the comparison of the values of protonation constants by NMR spectroscopy and glass-electrode potentiometry. The values used for data analysis have been taken from literature, mostly from author’s articles. Authors made nice SWOT analysis (see Table 1). There are some doubts and discrepancies mentioned here:

·       Novelty of paper is doubtful since authors made comparison literature values obtained by two experimental techniques (NMR spectroscopy vs. glass-electrode potentiometry). Why have not been other experimental techniques tested? For example, molecular absorption spectroscopy?   

·       Authors used different experimental data measured by NMR and potentiometry while pH values were obtained with glass electrode or with pH buffer. In case, when the OH- or H+ concentration are much higher than the concentration of analyte, the acidity/basicity of solution could be calculated. When the glass-electrode is suitable for pH region 2-12, the methodology using buffer solution or the calculated values are much precise therefore only pKa/pKb values in region 2-12 could be compared.

·       The impact of experimental conditions (e.g., temperature, ionic strength) on obtained pKa/pKb values should be discussed and therefore this aspect should be mentioned in paper.

·       In some cases, some values are given with the estimation of standard deviation, and some are not. How to distinguish among these values? One possibility is the weighing of the values as w_i = 1/var_i = 1/s_i^2 as it is done in some software applied for the calculation of equilibrium constants.

 For the reasons mentioned above, I could not recommend this paper to publish in International Journal of Molecular Sciences journal in this stage because of serious remarks mentioned above. 

Author Response

Comment 1:

Novelty of paper is doubtful since authors made comparison literature values obtained by two experimental techniques (NMR spectroscopy vs. glass-electrode potentiometry). Why have not been other experimental techniques tested? For example, molecular absorption spectroscopy?

Response 1:

During the inception of this work our goal was only to assess agreement between the two methods used in our research group (NMR and potentiometry). Later as the scope of our meta-analysis widened and includes now the works of all literature data available to us, we focused more on presenting a sound methodology for comparing two methods and assessing their agreement. More abundant methods could also be included (e.g. UV-pH titration), however that is outside the scope of our study currently, but is a promising direction for the future. Our aim was to present the statistical method of assessing agreement to the medicinal chemistry community.

Comment 2:

Authors used different experimental data measured by NMR and potentiometry while pH values were obtained with glass electrode or with pH buffer. In case, when the OH- or H+ concentration are much higher than the concentration of analyte, the acidity/basicity of solution could be calculated. When the glass-electrode is suitable for pH region 2-12, the methodology using buffer solution or the calculated values are much precise therefore only pKa/pKb values in region 2-12 could be compared.

Response 2:

The reviewer is correct in pointing out the issues with pH determination. However, the common practice of pH determination was established long ago and the data analyzed in this work contains works of recent papers and older ones as well. It is discussed that pKa determination is best evaluated in the mentioned range:

"With these insights in mind, NMR remains a powerful tool for deter-392 mining acid dissociation/protonation constants, even when high accuracy data are re-393 quired. In fact, outside the pKa range of 2-10 the use of NMR-titration with in situ pH 394 indicators is preferable. In the intermediate pKa range the two methods can be used inter-395 changeably."

Comment 3:

The impact of experimental conditions (e.g., temperature, ionic strength) on obtained pKa/pKb values should be discussed and therefore this aspect should be mentioned in paper.

Response 3:

The impact of experimental conditions is included in the Results section:

"In order to gauge the effect of measurement parameters (ionic strength, temperature) 245 the above mixed-effects models were extended in their fixed effects by these variables in 246 interaction with the titration method. No significant effects were observed for any inter-247 action terms, then based on standard model selection procedures the non-significant in-248 teraction terms followed by main effect terms were removed, until no significant effects 249 remained from ionic strength and temperature. Furthermore, in the Bland-Altman plot of 250 the trimmed data it was examined whether data points were affected by a differing tem-251 perature or ionic strength condition when determined with the two different titration 252 methods, however the number of such data was small and there was no observable trend 253 with respect to this comparison. These results can be reproduced using the R code in the 254 Supporting Information."

Comment 4:

In some cases, some values are given with the estimation of standard deviation, and some are not. How to distinguish among these values? One possibility is the weighing of the values as w_i = 1/var_i = 1/s_i^2 as it is done in some software applied for the calculation of equilibrium constants.

Response 4:

In the manuscript the following was added:

"Unfortunately, many instances of literature papers do not report the uncertainty of determination (standard error or standard deviation), however in the mixed-effects model the standard error of the values were not incorporated directly in the model, since it is apparent from the distribution standard errors that an appropriate variance weight function can be used, i.e. a lack of standard error was not considered as an exclusion criterion."

Unfortunately, not all literature data report standard errors. Although using the inverse of the variance is commonly used as weights for the calculation of the population mean in meta-analysis, our focus was on assessing agreement, not the mean of a certain parameter. Therefore the inclusion of the standard errors in the mixed-effects model was not necessary.

Round 2

Reviewer 2 Report

Comments and Suggestions for Authors

Authors accepted some my remarks, therefore it could be published in IJMS journal.